# MGCAF: A Novel Multigraph Cross-Attention Fusion Method for Traffic Speed Prediction

**DOI:** 10.3390/ijerph192114490

**Published:** 2022-11-04

**Authors:** Tian Ma, Xiaobao Wei, Shuai Liu, Yilong Ren

**Affiliations:** 1School of Automation Science and Engineering, Beihang University, Beijing 100191, China; 2School of Transportation Science and Engineering, Beihang University, Beijing 100191, China; 3Zhongguancun Laboratory, Beijing 100094, China; 4Beihang Hangzhou Innovation Institute Yuhang, Hangzhou 310023, China

**Keywords:** traffic speed prediction, graph convolutional network, cross-attention

## Abstract

Traffic speed prediction is an essential part of urban transportation systems that contributes to minimizing the environmental pollution caused by vehicle emissions. The existing traffic speed prediction studies have achieved good results, but some challenges remain. Most previously developed methods only account for road network characteristics such as distance while ignoring road directions and time patterns, resulting in lower traffic speed prediction accuracy. To address this issue, we propose a novel model that utilizes multigraph and cross-attention fusion (MGCAF) mechanisms for traffic speed prediction. We construct three graphs for distances, position relationships, and temporal correlations to adequately capture road network properties. Furthermore, to adaptively aggregate multigraph features, a multigraph attention mechanism is embedded into the network framework, enabling it to better connect the traffic features between the temporal and spatial domains. Experiments are performed on real-world datasets, and the results demonstrate that our method achieves positive performance and outperforms other baselines.

## 1. Introduction

Intelligent transportation systems (ITS) are important for reducing vehicle carbon emissions and providing environmental protection. An ITS can significantly inhibit CO_2_ emissions caused by transportation, but this depends on the development of ITS in the region [1]. Transportation in China results in the annual production of 14.76 million tons of CO_2_ emissions [2]. However, a 1% increase in an ITS as a percentage of the transportation system in one province can lead to CO_2_ emissions decreasing by 0.1572% locally and by 0.3535% in a neighboring province; this leads to a reduction of 0.3022% on average in China [3]. In short, an ITS has a significant negative impact on carbon emissions, as it decreases CO_2_ emissions by improving transportation structure and technology.

To build an ITS in cities, one of the most critical aspects is to predict traffic speed, which has a significant effect on daily life [4]. Accurate traffic speed prediction has a direct impact on urban traffic networks and travel planning. On the one hand, people can reasonably adjust their trip plans based on predicted traffic speeds and ease traffic congestion during peak hours. On the other hand, an appropriate vehicle speed reduces fuel consumption and greenhouse gas emissions [3]. Thus, traffic speed prediction is an essential component of an ITS.

The existing traffic speed prediction methods have obtained promising results on several datasets. However, some challenges remain. Since road networks are non-Euclidean networks with complex spatial–temporal correlations, a graph convolution network (GCN) was used to mine spatial dependencies, and a recurrent neural network (RNN) was used to learn the temporal correlations in previous works. With the development of GCN and RNN, many approaches can handle data derived from different aspects and achieve promising performance in terms of traffic speed prediction accuracy.

However, these methods still have several drawbacks. The majority of current works focus on limited road network characteristics, such as the distances between nodes, to represent the relationships between urban road nodes. An urban traffic network possesses various external properties, such as distances, road directions, and temporal patterns, which are highly related to traffic speed and density. Several approaches design multiple graphs to compensate for missing information and extract hidden features about the given road network for traffic speed prediction. Nevertheless, these works tend to model only the spatial dependencies of the studied road network and feed data into an RNN model to extract temporal features indirectly. For a traffic network, the connectivity between space and time is a key feature that cannot be ignored. For example, in an urban road network, traffic detection sensors in residential and commercial areas exhibit different results at different times. We believe that manually constructing temporal correlation graphs can simultaneously capture temporal and spatial characteristics and genuinely reflect the dynamically changing temporal patterns in different traffic districts.

In this paper, we propose a novel model that utilizes multigraph and cross-attention fusion (MGCAF) mechanisms to solve the shortcomings mentioned above and help reduce carbon emissions. To capture both complicated spatial dependencies and temporal correlations, we construct three graphs for distances, position relationships, and temporal correlations to take the actual characteristics between nodes and time series into account. Additionally, to further explore the correlation between the spatial and temporal domains, we apply an attention mechanism and a data fusion operation to fuse the multigraph features extracted by the GCN in our model. Then, we feed the spatial-temporal data into a GCN-RNN-based network and propose an attention-based feature fusion mechanism to thoroughly capture features and improve the accuracy of traffic speed prediction.

In short, the main contributions of our work are as follows.

We construct three graphs for distances, position relationships, and temporal correlations as the fundamental building blocks to directly capture the spatial–temporal correlations in road networks.We further utilize an attention mechanism and a data fusion operation to explore the connection between the spatial and temporal domains in spatial–temporal data. A traffic speed prediction network is proposed to thoroughly capture features in road networks.To conduct thorough comparisons and test the performance of our approach in complicated cases, experiments are conducted on several real-world datasets. The results prove that our proposed MGCAF model achieves positive performance and assists with the construction of ITS, which may lead to lower carbon emissions.

The rest of this paper is organized as follows. Section 2 reviews existing GCN- and RNN-based methods for traffic speed prediction. Section 3 gives the definition of the traffic speed prediction problem and details our proposed MGCAF method, including its multigraph construction mechanism, multigraph spatial–temporal feature extractor and multigraph feature cross-attention fusion technique. Section 4 introduces the experimental results, performance discussions and an ablation study. Section 5 is a conclusion of the whole article.

## 2. Related Works

Traffic speed prediction remains a challenging task due to the complex relationships and heterogeneity in the space and time domains. To solve this problem, a large number of datasets that contain adequate real-world traffic information have been proposed. In the preceding decades, numerous approaches for traffic speed prediction have been presented in an effort to improve prediction accuracy and resilience.

Early traffic speed prediction efforts relied heavily on statistical methodologies and machine learning techniques. Hamed et al. [5] first used the autoregressive integrated moving average (ARIMA) model for traffic speed prediction with city road datasets. Dauwels et al. [6] applied unsupervised learning methods. They adopted K-means, principal component analysis (PCA), and self-organizing maps to identify spatial and temporal performance patterns and used support vector regression (SVR) to make traffic speed predictions. To better capture the relations between sequential traffic flow data, Wang et al. [7] utilized a novel learning approach involving a relevance vector machine to achieve better traffic speed prediction performance. Chen et al. [8] introduced a modified k-nearest neighbors (KNN) model on the basis of spatial–temporal relationships to improve the performance of traffic speed predictions. In addition, the Kalman filter and its derivatives are commonly employed to solve traffic speed prediction problems [9]. Liu et al. [10] proposed a traffic state prediction model based on large-scale empirical data and obtained promising results.

For traffic speed prediction, models capable of simulating highly nonlinear temporal relationships are necessary due to the fast increases observed in computer power and the volume of traffic data over the last few years. Hua et al. [11] were the first to use a feedforward neural network for estimating traffic trip times. Subsequently, several other models based on deep learning were developed to estimate traffic conditions. RNNs and their variants (e.g., long short-term memory (LSTM) [12] and gated recurrent units (GRU) [13]) use a chain-like topology to represent the state space. Lint et al. [14] and Zhao et al. [15] used an RNN and LSTM, respectively, for freeway travel time prediction and short-term traffic speed prediction. Zhu et al. [16] proposed a dynamic traffic incident prediction approach based on LSTM and a multilayer perceptron (MLP). Convolutional neural networks (CNN) have advantages over RNN in terms of calculation efficiency. For example, Dai et al. [17] proposed a CNN-based approach that models traffic systems as 2D images and represents the spatial and temporal connectivity of traffic speed flow via a two-dimensional time-space matrix. Ma et al. [18] proposed a novel hybrid short-term traffic state prediction method based on critical road selection optimization. Ren et al. [19] proposed an LSTM-based method for short-term traffic prediction; their approach performs well in short-term traffic prediction cases involving special events. In addition, further approaches based on deep learning have been proposed for traffic speed prediction, such as deep belief networks (DBN) [20].

However, the aforementioned approaches are unable to fully exploit latent spatial relationships. To capture geographical and temporal relationships, ST-ResNet [21] first divides different traffic zones into grids according to their longitudes and latitudes. Then, it models the observed spatial and temporal dependencies using residual convolutional units and LSTM, respectively. CNN is unable to process non-Euclidean topologies between traffic nodes since it can only process conventional grid structures. Recently, graph neural networks (GNN) have been created to circumvent this constraint.

A GNN describes arbitrary graphs using an adjacency matrix so that it has the ability to process non-Euclidean-like data. Numerous studies have integrated GNN with learning-based time series networks to concurrently extract spatial and temporal relationships. Zhao et al. [22] utilized graph convolution to represent spatial dependencies and a GRU to represent temporal dependencies. The STGCN [23] comprises several spatiotemporal convolutional blocks that consist of ChebNet [24] and gated CNN. Additionally, the attention-based STGCN (ASTGCN) [25] was established as a spatial–temporal attention method to dynamically learn the relationships between time and space. In a similar fashion, Guo et al. [26] presented the graph attention-based temporal convolutional network (GATCN), which addresses spatial features via a graph attention network and temporal features with a TCN. Dai et al. [27] presented a spatial–temporal transformer network (STTN) consisting of a GNN and a transformer [28] to dynamically simulate different scales of geographical dependencies and capture long-range temporal connections. Lv et al. [29] used a message passing system and multiple kinds of LSTM to describe spatial–temporal relationships. Li et al. [30] aimed to solve online car-hailing problems with an ST-transformer model and analyzed the spatiotemporal characteristics of online car-hailing data. Shi et al. [31] applied an attention mechanism to a GCN-GRU-based network to capture the importance of different temporal features.

An adjacency matrix, which shows the correlations between nodes, is a crucial component of a GNN. Most studies create adjacency matrices based on the Euclidean distances between sensors for traffic speed prediction tasks. Chen et al. [32] noted, however, that the constructed graph disregards the intricacy and interplay of edges. To describe the correlations between edges, additional adjacency matrices corresponding to an edgewise graph were built. Predefined adjacency matrices cannot fully describe the topological structure of a traffic network, although the aforementioned approaches have achieved high performance. To overcome this issue, the AGCN [33] trains a task-driven adaptive graph for each task during training, making the spectral convolution kernel genuinely practical given the heterogeneous graph topology of the input data. Graph WaveNet [34] constructs a self-adaptive adjacency matrix using learnable source node embedding and destination node embedding. However, it presupposes that there is a hidden dependence between any two nodes, leading to redundant spatial correlations. The STFGNN designs a pipeline to integrate hidden temporal and spatial properties, which include local and global correlations [35]. The DMGCRN proposed a dynamic multigraph construction method to capture dynamic temporal and spatial properties [36]. The AGCRN also proposed an adaptive graph convolutional network to integrate dynamic features [37]. The DDP-GCN designs a multigraph construction mechanism to better represent road network, which helps the network learn the features of traffic systems [38].

However, compared with implicit feature extraction, explicit multigraph feature construction is more conducive to network learning. In addition, the use of a cross-attention mechanism allows the constructed network to learn more correlations between different features.

## 3. Methodology

### 3.1. Problem Definition

For a graph *G*, we use G=(V,E,A) to represent road networks, where *V* is the set of vertices that denote *N* real-world traffic sensor nodes (|V|=N), *E* is the set of edges in the graph that represents the roads in urban areas, and $A\in R^{N\times N}$ is the adjacency matrix for representing the correlations and connectivity between nodes. In general, the adjacency matrix contains information such as the directions and the distances between nodes (traffic sensors). Multiple graph signal-containing traffic series are represented as $\left\{X_{1},\cdots ,X_{t},\cdots \right\}$. At time step *t*, each graph signal is defined as $X_{t}\in R^{N\times d}$, where *d* is the input feature size. The corresponding feature is constructed manually or extracted by the network. For traffic speed prediction, our goal is to optimize a model f(·) to fit the structure of real road networks. The model takes traffic speed sequences of length *T* as inputs and gives predictions for the traffic speeds in the next $T^{\prime }$ time steps.
(1)$$\begin{array}{c} \left\{X_{t+1},\cdots ,X_{t+T^{\prime }}\right\}=f\left(X_{t-T+1},\cdots ,X_{t};G\right) \end{array}$$

### 3.2. MGCAF Framework

Figure 1 illustrates the architecture of our MGCAF model. We summarize our model based on its three core components: (1) a multigraph construction mechanism, (2) a spatial–temporal feature extractor, and (3) a cross-attention fusion module.

#### 3.2.1. Multigraph Construction Mechanism

In this work, we construct multigraphs on directed road networks for traffic speed prediction; these multigraphs include distance, direction and time graphs to explicitly represent the spatial and temporal correlations between nodes. As shown in Figure 2, we construct three kinds of adjacency matrices according to the topologies of road networks.

For the distance matrix $A_{s}$, we consider the shortest distance between 2 nodes *i* and *j* as D(i,j). It is not difficult to obtain such distances from datasets. When a distance is not given, we adopt Dijkstra algorithm [39] to estimate it. Thus, the adjacency matrix can be formulated as:(2)$$A_{s}\left(i,j\right)=\left\{\begin{array}{cc} D(i,j), & \;i\neq j\;\\ 0, & \;\mathrm{otherwise}\; \end{array}\right.$$

For the direction matrix $A_{d}$, we follow the construction method used in DDP-GCN [38] and define $A_{d}=\left\{A_{d}^{1},A_{d}^{2},A_{d}^{3},A_{d}^{4}\right\}$. Since the directions between nodes belong to four cases, we can obtain four adjacency matrices. For each case, if there is a directed link between the two nodes *i* and *j*, $A_{d}^{k}\left(i,j\right)$ is set to 1. This idea can be formulated as:(3)$$A_{d}\left(i,j\right)=\left\{\begin{array}{cc} 1, & \;\mathrm{if}\;\mathrm{node}\;i\;\mathrm{connects}\;\mathrm{to}\;j\;\mathrm{in}\;\mathrm{one}\;\mathrm{direction}\\ 0, & \;\mathrm{otherwise}\; \end{array}\right.$$

For the time matrix $A_{d}$ that represents the temporal correlations between nodes, we consider the connectivity of a node *i* with its neighboring node *j* at adjacent moments. When node *i* connects to node *j* at adjacent moments, we set $A_{d}\left(i,j\right)$ to 1. Thus, we can obtain a sequence of adjacency matrices. The diagonal lines of time matrices represent the connectivity of nodes at adjacent moments, while the nondiagonal regions represent the connectivity of neighboring nodes. Finally, each element can be formulated as:(4)$$A_{t}\left(i,j\right)=\left\{\begin{array}{cc} 1, & \;\mathrm{if}\;\mathrm{node}\;i\;\mathrm{connects}\;\mathrm{to}\;j\;\mathrm{at}\;\mathrm{adjacent}\;\mathrm{moments}\\ 0, & \;\mathrm{otherwise}\; \end{array}\right.$$

#### 3.2.2. Concurrent Spatial-Temporal Feature Extractor

As seen in Figure 1, the MGCAF model incorporates *N* spatial–temporal feature extractors to concurrently capture spatial and temporal relationships. A spatial–temporal feature extractor comprises a spatial graph convolutional layer (GCN) and a gated recurrent unit (GRU).

For better feature representation learning, skip connections are used between layers of the same kind to communicate spatial and temporal characteristics. The calculation proceeds as follows:(5)$$\begin{array}{ccc} H_{t}^{n} & = & F_{t}(H_{s}^{n-1}+\varphi_{t}\left(H_{t}^{n-1}\right))\;\;\; \end{array}$$(6)$$\begin{array}{ccc} H_{s}^{n} & = & F_{s}\left(H_{t}^{n}+\varphi_{s}\left(H_{s}^{n-1}\right),G\right) \end{array}$$
where $H_{t}^{n},H_{s}^{n}\in R^{B\times T\times D}$ are the output temporal and spatial features of layer n,n∈{1,⋯,N}, respectively. $\varphi_{t}\left(\cdot \right)$ and $\varphi_{s}\left(\cdot \right)$ are linear transformations. $F_{t}\left(\cdot \right)$ and $F_{s}\left(\cdot \right)$ represent the temporal feature extractor and spatial graph feature extractor, respectively. *G* denotes the multigraphs constructed in Section 3.2.1.

(1) Temporal Feature Extractor

We use a convolutional neural network (CNN) to simulate the temporal dependence because it fully exploits parallel processing and creates a steady gradient. Residual connections are used to solve the network deterioration issue encountered in deep neural networks. In addition, we use a gating technique to regulate the flow of information between layers.
(7)$$\begin{array}{ccc} G_{t}^{n-1} & = & \sigma (\Gamma_{\Theta_{1}}\left(H_{t}^{n-1}\right))\; \end{array}$$
(8)$$\begin{array}{ccc} {\tilde{H}}_{t}^{n-1} & = & \phi (\Gamma_{\Theta_{2}}\left(H_{t}^{n-1}\right))\; \end{array}$$
(9)$$\begin{array}{ccc} \;\;H_{t}^{n} & = & G_{t}^{n-1}⊗{\tilde{H}}_{t}^{n-1}+H_{t}^{n-1}\;\;\; \end{array}$$
where σ(·) and ϕ(·) are the sigmoid and tanh functions that control the features conveyed to the next layer. $\Gamma_{\Theta }\left(\cdot \right)$ is a one-dimensional convolution operation with a parameter Θ. ⊗ represents the Hadamard product.

(2) Spatial Graph Feature Extractor

A graph convolutional process is a better way to extract features from each node and its neighbor nodes in road networks. Based on the three kinds of constructed graphs, we adopt the GCN operation to aggregate the graph features at different time steps. For each GCN layer, the described process can be formulated as follows:(10)$$H_{s}^{n}=GCN\left(H_{s}^{n-1}\right)=\sigma \left(A^{\prime }H_{s}^{n-1}W+b\right)$$
where σ(·) is the activation function, such as a rectified linear unit (ReLU) or gated linear unit (GLU) function [40]. $H_{s}^{n-1}$ denotes the input of the n-th GCN layer. $A^{\prime }$ represents the three kinds of adjacency matrices constructed above. *W* and *b* denote learnable parameters in the networks. To achieve a better feature extractor, we choose a GLU as our activation method; thus, the formulation above can be rewritten as follows:(11)$$H_{s}^{n}=\left(A^{\prime }H_{s}^{n-1}W_{1}+b_{1}\right)⊗\sigma \left(A^{\prime }H_{s}^{n-1}W_{2}+b_{2}\right)$$
where σ(·) denotes the sigmoid function. $W_{1}$, $b_{1}$, $W_{2}$, and $b_{2}$ are learnable parameters. ⊗ denotes the elementwise product operation.

#### 3.2.3. Graph Feature Cross-Attention Fusion

Our proposed MGCAF model utilizes the features derived from all the feature extractors to compute the speed predictions for the final $T^{\prime }$ timesteps as outputs. The model realizes direct and early supervision for all the feature extractors, leading to faster convergence and better performance. For each graph convolution block, we first apply self-attention modules to refine the features extracted from the constructed graph. We can formulate the self-attention modules as follows:(12)$${\tilde{H}}_{h}=\mathrm{Softmax}\left(\frac{\left(HW_{h}^{Q}\right){\left(HW_{h}^{K}\right)}^{T}}{\sqrt{d_{k}}}\right)\left(HW_{h}^{V}\right)$$
where $W_{h}^{K}\in R^{D\times d_{k}}$ and $W_{h}^{Q}\in R^{D\times d_{k}}$ are transformation matrices for queries and keys of size $d_{k}$. $W_{h}^{V}\in R^{D\times d_{v}}$ is also a transformation matrix for the value vector of size $d_{v}$. They are all learnable parameters. Then, multiple cross-attention layers are utilized to fuse the features derived from different graphs, which helps the network learn the spatial–temporal correlations between nodes.
(13)$${\tilde{H}}_{h}^{\prime }=\mathrm{Softmax}\left(\frac{\left({\tilde{H}}_{h}^{d}W_{h}^{Q}\right){\left({\tilde{H}}_{h}^{s}W_{h}^{K}\right)}^{T}}{\sqrt{d_{k}}}\right)\left({\tilde{H}}_{h}^{s}W_{h}^{V}\right)$$
(14)$${\tilde{H}}_{h}=\mathrm{Softmax}\left(\frac{\left({\tilde{H}}_{h}^{t}W_{h}^{Q}\right){\left({\tilde{H}}_{h}^{\prime }W_{h}^{K}\right)}^{T}}{\sqrt{d_{k}}}\right)\left({\tilde{H}}_{h}^{\prime }W_{h}^{V}\right)$$
where $W_{h}^{K}$, $W_{h}^{Q}$ and $W_{h}^{V}$ are all learnable parameters. ${\tilde{H}}_{h}^{d}$, ${\tilde{H}}_{h}^{s}$ and ${\tilde{H}}_{h}^{t}$ indicate features that were refined after passing through the self-attention layers of the direction, distance and time graph convolution blocks, respectively. The direction features are first used as the query and fused with the distance features. Then, the spatial fusion results are fused with the time features as the final fusion results. Finally, we concatenate all features obtained from the spatial–temporal feature extractors as the input of the global attention layer. Followed by a fully connected layer, the MGCAF model finally predicts the $T^{\prime }$-step speed sequence $\tilde{Y}\in R^{N\times T^{\prime }\times 1}$ as the model output.

## 4. Experiments

### 4.1. Datasets and Preprocessing

We assess our model using two common datasets for traffic speed forecasting, METR and PeMS, which have been derived from the actual world. (1) The Performance Measurement System (PeMS) was gathered by California Transportation Agencies (CalTrans) in the Bay Area of California, which captured six months of traffic speed data from 1 January 2017 to 31 May 2017, with 325 sensors. (2) METR was gathered on a Los Angeles County roadway. From 1 March 2012 to 30 June 2012, 207 sensors captured four months of traffic velocity data. The statistical overview is shown in Table 1.

The preprocessing method adheres to the diffusion convolutional RNN (DCRNN) [41]. The datasets are proportionately divided into training, validation, and test sets. 70% of the data are used for the training set, 30% of the data are used for the validation set, and 10% of the data are used for the test set.

### 4.2. Implementation Details

Our proposed MGCAF model takes the previous 12-speed sequence as input and predicts the speed sequence for the next 12 timesteps. The model consists of 8 spatial–temporal feature extractors. For the attention layers, we apply 8 self-attention layers, 8 cross-attention layers and 4 global attention layers. We adopt the Adam optimizer with an initial learning rate of 0.001 and a weight decay of 0.001 to train our model. We train for 100 epochs on both PeMS and METR with a batch size of 32.

### 4.3. Performance Analysis

To demonstrate the high accuracy of the proposed method, we compare the MGCAF inference results to those of the following techniques.

(1) FNN: Feedforward Neural Network; (2) FC-LSTM: Fully Connected LSTM; (3) STGCN [23]: Spatial-Temporal Graph Convolutional Network; (4) DCRNN [41]: Diffusion Convolutional Recurrent Neural Network; (5) STGNN [42]: Spatial–Temporal Graph Neural Network; (6) STFGNN [35]: Spatial–Temporal Fusion Graph Neural Network. (7) DMGCRN [36]: Spatial–Temporal Fusion Graph Neural Network. (8) DDP-GCN [38]: Spatial–Temporal Fusion Graph Neural Network.

The mean absolute error (MAE), root mean square error (RMSE), and mean absolute percentage error (MAPE) are used as the metrics in accordance with past research. During the metric computation process, missing values are omitted from the datasets. For timestep *t*, the following metrics are computed:(15)$$\mathrm{MAE}\left(\hat{Y},Y\right)=\frac{1}{N}\sum_{i=1}^{N}\left|{\hat{Y}}_{t}^{\left(i\right)}-Y_{t}^{\left(i\right)}\right|$$
(16)$$\mathrm{RMSE}\left(\hat{Y},Y\right)=\sqrt{\frac{1}{N}\sum_{i=1}^{N}{\left({\hat{Y}}_{t}^{\left(i\right)}-Y_{t}^{\left(i\right)}\right)}^{2}}$$
(17)$$\mathrm{MAPE}\left(\hat{Y},Y\right)=\frac{1}{N}\sum_{i=1}^{N}\left|\frac{{\hat{Y}}_{t}^{\left(i\right)}-Y_{t}^{\left(i\right)}}{Y_{t}^{\left(i\right)}}\right|$$
where *N* denotes the number of samples and ${\hat{Y}}_{t}^{\left(i\right)}$ and $Y_{t}^{\left(i\right)}$ denote the ground-truth speed and the predicted speed for timestep *t*, respectively.

Table 2 shows the experimental results obtained by our MGCAF model and other baseline methods on the PeMS and METR datasets for three different timesteps.

From the tables, we observe the following. (1) Our proposed MGCAF model outperforms the FNN and FC-LSTM with reduced inference error because the latter two methods can only extract temporal information and disregard spatial dependencies, therefore losing a large amount of information and causing significant performance degradation. (2) As indicated in the tables, our method achieves competitive results in comparison with those of the spatiotemporal GCNs in earlier techniques, such as the STGCN, DCRNN, and STFGNN. This demonstrates the efficacy of the multigraph fusion and cross-attention mechanism in the time and space domains. (3) In contrast to the fusion method suggested in the STFGNN, we use a cross-attention strategy that enables the network to integrate the connections between temporal and spatial characteristics, resulting in decreased prediction error rates on the two test sets. (4) In contrast to the recent multigraph construction method suggested in the DMGCRN and DDP-GCN, our method still performs better in most of metrics.

In Figure 3, the actual and predicted speeds obtained for the test subgroups of PeMS and METR are shown. Our model fits the training subsets well and is generally immune to noise. The traffic speed in the first subfigure (row 1, column 1) fluctuates substantially. The anticipated curve nearly matches the real curvature; therefore, our model can handle this situation successfully. In row 2, column 2 of Figure 3, a negligible traffic speed variance is observed. In this instance, which requires the ability to capture sequence information and features that are helpful for prediction, our model also performs well and almost fits the real speed curve. In row 3, column 2 of Figure 3, missing values have been replaced with zeros. Here, our model delivers accurate predictions, illustrating its resistance and robustness to outliers.

### 4.4. Ablation Study

We conduct experiments to confirm the contributions of the improvement’s essential components. We remove the multigraph construction mechanism and cross-attention fusion module from the MGCAF model. Then, it degenerates into a general STGNN. Finally, we individually apply the multigraph construction mechanism and cross-attention fusion module to the STGNN to compare the obtained testing results.

The experiments are carried out using the exact same conditions as those employed during training. The experimental results are shown in Table 3.

The MGCAF model (row 4 in Table 3) demonstrates the highest performance. Based on the findings shown in rows 1 and 2 of Table 3, the efficacy of the multigraph construction mechanism is determined. This shows that single-graph construction process has a negative impact on performance due to the limited obtained features. This is because multigraphs that contain distances, position relationships, and temporal correlations are more similar to real road networks. Comparing rows 2, 3, and 4 in Table 3, it is discovered that the achieved performance is enhanced when attention layers are included, and when vanilla attention evolves into cross-attention, the resulting performance also improves. Due to the noise and missing values in datasets, the attention mechanism is necessary to denoise and fully fuse the features derived from different graphs.

## 5. Conclusions

To reduce the carbon emissions produced by transportation and alleviate the pressure imposed on urban traffic systems, we propose a model for increasing the accuracy of traffic speed prediction, which can hasten the development of ITS. Existing traffic speed prediction approaches have achieved good results, but still have several drawbacks. In this study, we introduce a novel network named MGCAF. The MGCAF network is a multigraph and cross-attention fusion network for predicting traffic speed. By utilizing multigraph structures, the model captures numerous aspects of road networks in the actual world. On the basis of the three constructed graphs, we use an attention mechanism and data fusion operation to explore the relations between the spatial and temporal domains. The model provides substantial enhancements for two real-world datasets. An ablation study further reveals that the developed multigraph fusion mechanism contributes significantly to the model. In the future, we want to construct more precise models for road networks, which may increase traffic speed prediction accuracy and lead to lower carbon emissions.

## Figures and Tables

**Figure 1 ijerph-19-14490-f001:**
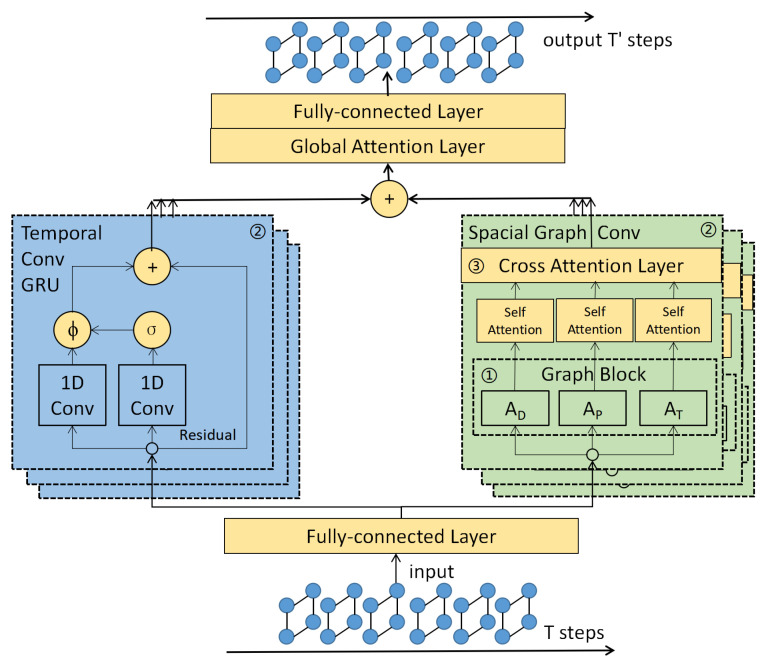
The overall MGCAF framework. Our proposed MGCAF model consists of (1) a multigraph construction mechanism, which consists of distances, position relationships, and temporal correlations as the fundamental building blocks. We use Graph Block to represent the mechanism in the figure; (2) a spatial–temporal feature extractor, which is formed of several GRU and GCN modules to extract spatial–temporal features simultaneously. We use blue and green colors to represent the blocks; and (3) a cross-attention fusion layer is used to fuse the features extracted by the GCN modules. The cross-attention fusion layer is located in the spatial feature extractors, which are represented in yellow in the figure.

**Figure 2 ijerph-19-14490-f002:**
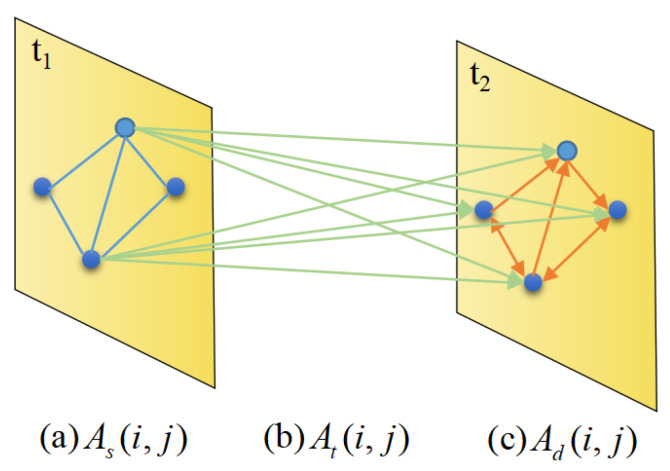
The three kinds of adjacency matrices that we construct. (**a**) $A_{s}$ indicates the shortest path distances between nodes, which are indicated as blue lines, (**b**) $A_{d}$ denotes the directions between different nodes, which are indicated as orange lines, and (**c**) $A_{t}$ represents the temporal correlations between nodes and their neighbors at adjacent moments, which are indicated as green lines.

**Figure 3 ijerph-19-14490-f003:**
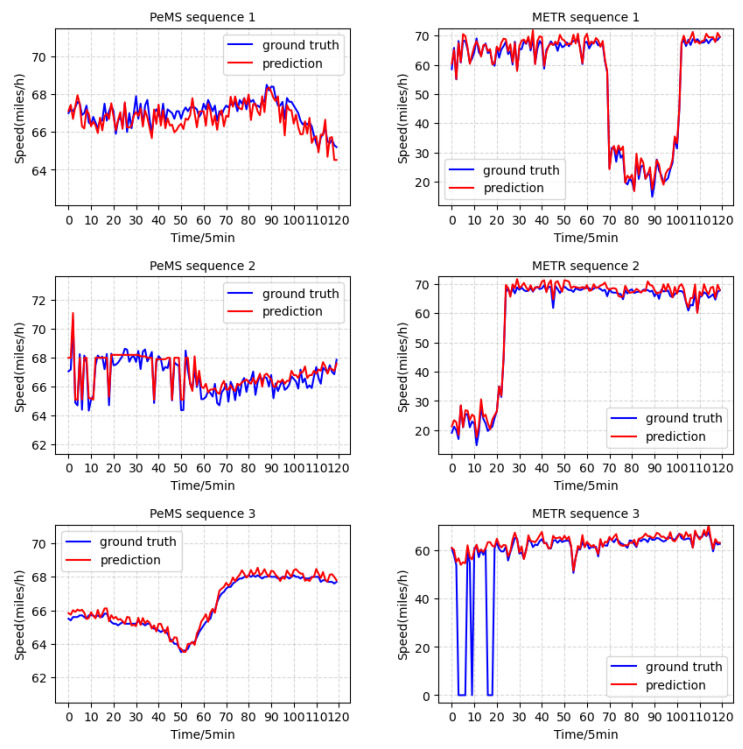
Visualization of the ground-truth speed sequence and the prediction results obtained on the PeMS and METR datasets.

**Table 1 ijerph-19-14490-t001:** Summary statistics of two real world datasets. We count the numbers of nodes, edges and timesteps in both the PeMS and METR datasets and organize them into tables.

Dataset	#Nodes	#Edges	#Timesteps
PeMS	325	2369	52116
METR	207	1515	34272

**Table 2 ijerph-19-14490-t002:** The inference performance achieved by the MGCAF model and different baselines on two datasets.

Data	Models	15 min (t = 3)			30 min (t = 6)			60 min (t = 12)		
		MAE	RMSE	MAPE	MAE	RMSE	MAPE	MAE	RMSE	MAPE
PeMS	FNN	3.24	5.42	6.12%	2.65	4.25	5.14%	2.14	4.98	5.43%
	FC-LSTM	2.29	4.36	4.96%	2.36	4.17	5.02%	2.29	5.01	5.26%
	STGCN	1.35	2.58	2.95%	1.24	4.14	4.02%	2.98	5.14	5.20%
	DCRNN	1.32	2.94	2.96%	1.85	3.97	3.96%	2.06	4.86	4.96%
	STGNN	1.31	2.98	2.95%	1.86	3.96	3.93%	2.68	4.82	4.86%
	STFGNN	1.30	2.83	2.90%	1.84	3.91	3.90%	2.02	4.73	4.75%
	DMGCRN	**1.25**	2.80	2.91%	1.80	3.90	3.91%	2.00	4.71	4.74%
	DDP-GCN	1.28	**2.78**	2.88%	1.76	3.92	3.86%	1.98	4.69	4.74%
	MGCAF	1.30	2.80	**2.88**%	**1.72**	**3.89**	**3.85**%	**1.96**	**4.68**	**4.72**%
METR	FNN	3.65	7.25	9.80%	4.25	8.65	13.52%	4.25	8.96	13.25%
	FC-LSTM	3.14	6.58	9.47%	3.69	7.21	10.76%	4.38	8.14	13.02%
	STGCN	2.68	5.72	8.25%	3.25	7.35	9.28%	4.69	9.14	12.25%
	DCRNN	2.87	5.36	7.21%	3.36	6.58	8.86%	3.58	7.60	10.60%
	STGNN	2.86	5.58	7.25%	3.36	6.47	8.82%	3.56	7.57	10.55%
	STFGNN	2.83	5.26	7.18%	3.27	6.38	8.73%	3.47	7.52	10.50%
	DMGCRN	2.61	5.01	6.72%	2.99	**6.04**	8.20%	3.38	**7.07**	**9.91**%
	DDP-GCN	2.65	5.12	6.73%	2.96	6.05	8.15%	3.36	7.10	9.93%
	MGCAF	**2.58**	**4.97**	**6.72**%	**2.80**	6.09	**8.11**%	**3.35**	7.13	10.11%

**Table 3 ijerph-19-14490-t003:** Ablation study. STGNN: a general spatiotemporal graph neural network; MG: using the multigraph construction mechanism; VA: using the vanilla attention layer for feature fusion; CA: using the cross-attention layer for feature fusion.

Model	PeMS			METR		
	Mean MAE	Mean RMSE	Mean MAPE	Mean MAE	Mean RMSE	Mean MAPE
STGNN	1.95	3.92	3.91%	3.26	6.54	8.87%
STGNN + MG	1.83	3.86	3.88%	3.21	6.46	8.32%
STGNN + MG + VA	1.70	3.84	3.83%	3.12	6.21	8.01%
STGNN + MG + CA(MGCAF)	**1.66**	**3.79**	**3.81**%	**3.04**	**6.05**	**7.91**%

## Data Availability

The data presented in this study are available on request from the corresponding author.

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
