# Peer review of "MGCAF: A Novel Multigraph Cross-Attention Fusion Method for Traffic Speed Prediction"

_ijerph, 2022, doi:10.3390/ijerph192114490_

Round 1

Reviewer 1 Report

The title corresponds to the content, the order and structure of the work is correct.
Minor editorial corrections are needed  
1. Line 174 the authors probably mean Figure 1 not Figure 2
2. Figure 1 in the caption is described that MGCAF model consists of 1..., 2..., 3... the numbers referred to in the description are not in the figure.
3. The first line below equation 3 describes that Ad represents the temporal correlations between nodes which does not agree with the description given in Figure 2
4. Lines 185 and 186 suggest that the graph convolutional layer is abbreviated GNN and the temporal convolutional layer is abbreviated GRU
5. Line 287 probably refers to column two and the third row.
Furthermore, are the authors sure of the statement in line 97 that Kalman Filter is nonparametric method.

Reviewer 2 Report

Dear sir

The paper is sound so good but I only suggest to change the title and make it more relevant to the paper in my view.

Reviewer 3 Report

This manuscript targets a multigraph and cross-attention fusion framework for traffic speed prediction to reduce carbon emissions. Especially, the authors construct three graphs for distances, position relationships, and temporal correlations to capture both complicated spatial dependencies and temporal correlations. Overall, this manuscript is well-organized and well-written in general, however, I have several concerns listed below before its potential acceptance.

(1) Related work and state-of-the-art, the reviewer noticed that most of the references were published before 2020, as for the GCN-CNN-based framework, it might lack the latest work in this field.

(2) 3.2 MGCAF Framework, which is too short as a subsection, the authors could supply related content to introduce each component.

(3) The configuration details and related parameters for the proposed work and comparative methods were missing.

(4) It is unclear how this research is explicitly linked to carbon emission reduction, also, it lacks a detailed discussion of the results and potential implications.

Round 2

Reviewer 3 Report

Thank you for addressing my concerns, however, I still have remaining concerns before its publication:

(1) Section 3.3~Section 3.5 could be sub-contents of Section 3.2, thus I will suggest relisting Section 3.3~Section 3.5 as Section 3.2.1~Section 3.2.3.

(2) The results do not discuss its direct connections to the carbon emissions reduction on urban traffic systems quantitatively, in this case, I will suggest the authors supply related analysis or just alter the expression regarding the carbon emissions reduction in the manuscript’s title since it only talks about the speed prediction.
